# CataRAG: LLM-based Enzyme-Substrate Prediction with Multi-modal Optimal Transport Maps

## Abstract

The prediction of catalytic interactions between enzymes and their substrates forms the foundation of numerous biological and medical processes, which, however comes with the challenge of inherent multi-modality of the biological information. Most link prediction methods are unimodal, and remain inadequate for complex biochemical reasoning and the integration of heterogeneous biological modality. Meanwhile, although current protein language models perform well in amino acid sequences embedding, they fall short in reasoning. To tackle these issues, we propose **CataRAG**, the first retrieval-augmented prediction biological reasoning framework, which builds on the multi-modal optimal transport theory to systematically integrate three key types of modalities, i.e., amino acid sequences, molecular structures of enzymatic substrates, and biological knowledge graphs, and connect heterogeneous biological data representations. Then the weighted average of the final retrieval results from heterogeneous sources can augment complex biological reasoning capabilities of LLMs without the need of retraining. Extensive experiments demonstrate on catalytic reaction prediction tasks that CataRAG significantly outperforms existing state-of-the-art LLMs, RAG baselines, and graph neural network models, with the crucial role of three modalities validated in ablation studies.

## 1 Introduction

Catalytic interactions between Enzymes and their substrates form the foundation of key processes in biology and medicine. These interactions support various biological functions from metabolism to signal transduction and are of significant importance to drug discovery and therapeutic development (Jumper et al., 2021; Zhang et al., 2023). Accurate prediction and understanding of these complex biochemical relationships remains a major challenge in computational biology, primarily because these interactions involve complex structural features, molecular dynamics, and multilevel biological regulatory mechanisms (Gainza et al., 2020). To enable effective reasoning and prediction, substantial efforts have been devoted to representing biological knowledge as knowledge graphs(Nicholson & Greene, 2022; Zeng et al., 2022), followed by applying graph learning methods such as graph neural networks (GNN) to perform tasks such as link prediction. However, these traditional approaches lack deep reasoning capabilities and exhibit poor generalization when dealing with unseen entities in the graph.

Large language models (LLMs), on the other hand, have demonstrated strong reasoning and generalization abilities. Yet, directly addressing biological tasks such as enzyme–substrate prediction in a question-answering format with LLMs often suffers from the problem of "hallucination": producing content that appears plausible but is factually incorrect(Brown et al., 2020; Chowdhery et al., 2022; Ji et al., 2023). A key reason for such hallucinations is the lack of domain-specific knowledge(Madani et al., 2023; Bommasani et al., 2022) or the inability to associate queries with the model's existing knowledge. To address this challenge, many studies have focused on pretraining the encoder component of large models with biological domain knowledge (Ferruz et al., 2022; Rives et al., 2021). For instance, some works have developed specialized language models to generate high-quality semantic embeddings from protein amino acid sequences (Lin et al., 2023b). However, such approaches

generally require substantial computational resources and remain limited in their applicability to complex biological prediction tasks (Burkart et al., 2022).

In this work, we propose CataRAG, a retrieval-augmented generation (RAG) framework that equips LLMs with biochemical reasoning capabilities. We leverage biological knowledge graphs and the UniProt database[1] as retrieval corpora, integrating domain-specific knowledge graph information, protein amino acid sequence structures, and catalytic molecular graph data to retrieve the most relevant molecular structures for catalytic relationship prediction. Biological knowledge graphs are particularly effective for representing the relational and structural organization of proteins, capturing complex associations such as genetic links, family hierarchies, and catalytic interactions.

To effectively integrate these heterogeneous sources of information—from graph-structured protein relations to sequence-level representations and molecular structures—we propose a multimodal optimal transport (OT) network to achieve semantic alignment across modalities. Specifically, this network, grounded in the Wasserstein distance, learns an optimal transport matrix mapping protein representation space to molecular representation space, thereby enabling robust cross-modal semantic alignment and similarity measurement. Amino acid sequences encode sequence and structural–evolutionary signals; OT preserves the global geometry of embedding spaces during alignment, a property equally essential for graph-based embeddings. Concretely, we learn an optimal transport plan between protein and molecular representation spaces to facilitate reliable cross-modal similarity assessment. Building on this, we adopt a three-pathway information fusion strategy.

The first is *graph structure information extraction*, where we use GNN to encode the neighborhood information of proteins in biological knowledge graphs, thereby capturing protein functional domains, interaction networks, and regulatory relationships. The second is *sequence information modeling*. Based on amino acid sequences from the UniProt database, we employ the pre-trained ProtBERT model to generate protein sequence embedding representations, preserving evolutionary and structural information. The third is *molecular structure representation*. For catalytic substrate molecules, we perform SMILES-to-Graph conversion and use molecular graph convolutional networks, to conduct comparative experiments with visual encoders for the evaluation of the performance differences between 2D molecular image representations and 1D string encodings. Following a weighted integration of these three representations, we use a learned ranking function to rank candidate molecules by relevance, retrieve the $k$ most relevant catalytic molecules along with their knowledge graph contexts, and finally input them into LLM for catalytic mechanism reasoning and biological explanation generation.

By equipping the LLM with structured knowledge from graphs, sequence embeddings, and molecular representations, our framework effectively bridges the knowledge gap that traditional LLMs face in biochemical reasoning. Moreover, because the retrieval and transport-based fusion operate on existing knowledge resources rather than training a domain-specific model from scratch, our approach avoids the prohibitive computational cost of retraining large-scale biological models. Overall, the proposed CataRAG enables large language models to integrate and analyze complex biochemical information from multiple data sources, substantially improving both the accuracy and reliability of catalytic interaction predictions.

The contribution of our work can be summarized as follows:

- **Multi-modal biochemical integration.** We design a Wasserstein optimal transport mechanism that couples protein sequences, molecular structures, and knowledge graphs, injecting biochemical knowledge into LLMs to fill their knowledge gap without costly biological PLM retraining.

- **Catalytic reasoning with LLM+RAG.** We propose *CataRAG*, the first retrieval-augmented framework for enzyme–substrate catalytic reasoning, where connecting multimodal biological information allows LLMs to reason with richer evidence for more accurate and explainable predictions.

- **Empirical validation.** Extensive experiments demonstrate that CataRAG significantly outperforms existing baselines in catalytic prediction tasks, achieving state-of-the-art performance. Ablation studies further validate the contribution of our retrieval enhancement

---

[1]https://www.uniprot.org/help/downloads

through the integration of these three modalities, confirming the necessity of multimodal retrieval and OT-based fusion for reliable biochemical reasoning.

Building on these contributions, our work introduces a novel paradigm for empowering foundation models towards biological discovery, demonstrating the potential of LLMs in structured reasoning within complex biological systems.

## 2 RELATED WORKS

In recent years, LLMs in the protein domain have made significant progress, yet their downstream applications remain limited. This section reviews related research, exploring the capabilities and limitations of existing models, as well as the potential of multi-modal retrieval-augmented generation in biochemical reasoning.

**Protein representation learning models** Protein-specific pre-trained models have achieved important breakthroughs in sequence representation. Models such as ESM (Evolutionary Scale Modeling) (Lin et al., 2023b; Rives et al., 2021), ProtBERT (Brandes et al., 2022), and ProtT5 (Elnaggar et al., 2021) learn rich evolutionary information and structural knowledge from vast protein sequences through self-supervised learning. These models effectively capture long-range dependencies and functional patterns in amino acid sequences through pretraining strategies like masked language modeling and sequence contrastive learning. In particular, models such as ESM-2 and ESM-Fold (Lin et al., 2023a) have demonstrated the ability to predict protein structures directly from sequences, providing a foundation for understanding protein function.

**Neural optimal transport and multi-modal fusion** Neural optimal transport (Korotin et al., 2022) approximates OT mappings through neural networks, which achieves effective alignment between different modalities. OT-Fusion (Singh & Jaggi, 2019) utilizes optimal transport theory to fuse visual and textual features, achieving excellent performance in multi-modal tasks. OT methods has also attracted attention in bioinformatics. Single-cell optimal transport (Schiebinger et al., 2019) applies optimal transport to single-cell data analysis to effectively solve cell trajectory inference problems. However, existing methods have not applied OT theory to protein catalytic prediction and focus on single biological data types, and thus lack a unified optimal transport framework for multi-modal biochemical data.

**Retrieval-augmented generation in scientific discovery** Retrieval-augmented generation (RAG) techniques have recently shown enormous potential in scientific domains. Models such as KG-RAG (Sanmartin, 2024) and BiomedRAG (Li et al., 2024) enhance generative capabilities by retrieving relevant scientific literature and knowledge graphs, while BioRAG (Wang et al., 2024) uses databases of biomedical literature for knowledge retrieval. However, these methods are primarily limited to a single modality (usually text), with no ability to integrate protein sequences, molecular properties, and knowledge graphs.

Despite significant advances in protein language models, their applications in complex downstream tasks remain limited. Key research gaps include: (1) lack of a unified framework that effectively integrates protein sequences, molecular properties, and knowledge graphs; (2) the difficulty of existing models in handling multi-modal biochemical data; and (3) the absence of specially designed reasoning frameworks for downstream tasks such as catalytic prediction. In contrast, CataRAG overcomes many of the limitations of related fusion approaches. CataRAG reasonably integrates the image information of the protein molecular formula, the information about neighboring nodes in the knowledge graphs, and the amino acid sequence information of the proteins as reasoning evidence to perform the link prediction and provide rational biological explanations.

## 3 PROPOSED FRAMEWORK: CATARAG

We propose a comprehensive protein representation learning framework that integrates multiple heterogeneous information sources. Our approach leverages three complementary modalities to capture the multi-faceted nature of protein functionality.

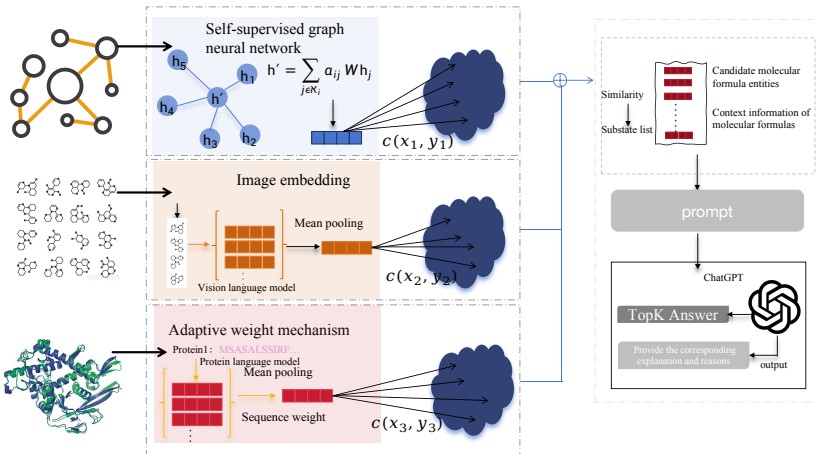

Figure 1: The figure illustrates the architecture of our multimodal catalytic relationship prediction RAG framework. On the left, three core feature extraction mechanisms are shown: (1) GNN process protein interaction networks to generate structured representations; (2) a VLM encoder embeds molecular structure via mean pooling; (3) a protein sequence encoder extracts weighted sequence features. They serve as three modules for protein representation. For each module, we design an optimal transport neural network to map protein embeddings to enzyme substrate embeddings. When we determine whether enzyme substrate x has a catalytic effect on protein y, we calculate the optimal transport distance for each module, then obtain the final similarity through weighted aggregation. We construct prompts by mining context information of candidate entities in the knowledge graph and feed them to the ChatGPT to generate final answers and explanatory information.

## 3.1 THREE PATHWAYS

**Knowledge graph representation.** We process biological knowledge graphs $\mathcal{G} = (\mathcal{V}, \mathcal{E}, \mathcal{R})$ using Graph Neural Networks (GNN). The message passing mechanism aggregates information from neighboring nodes through relation-specific transformations:

$$\mathbf{h}_i^{(l+1)} = \sigma \left( \mathbf{W}_{self}^{(l)} \mathbf{h}_i^{(l)} + \sum_{r \in \mathcal{R}} \sum_{j \in \mathcal{N}_i^r} \mathbf{W}_r^{(l)} \mathbf{h}_j^{(l)} \right) \tag{1}$$

where $\mathbf{W}_{self}^{(l)}$ and $\mathbf{W}_r^{(l)}$ are learnable layer-specific and relation-specific transformation matrices, $\mathcal{N}_i^r$ denotes the set of neighbors of node $i$ connected via relation $r$, and $\sigma$ is the activation function. This generates the knowledge graph embedding $\mathbf{h}_{kg}$ that captures rich relational context.

We leverage GNN over traditional knowledge graph embedding methods (e.g., TransE, ComplEx) due to GNN's superior ability to aggregate neighborhood representations through message passing. Unlike static embedding approaches that learn fixed representations, GNN dynamically aggregates structural and relational information from local neighborhoods, which is crucial for enhancing protein and molecular representations. For proteins and molecules with catalytic relationships, their functional similarity can be significantly enhanced through GNN's neighborhood aggregation mechanism, as functionally related entities tend to cluster in the biological knowledge graph topology.

**Sequence-based representation.** We extract protein amino acid sequences from the UniProt database and employ the AIDO.Protein-16B language model for sequence encoding, chosen over alternatives like ESM-2 or ProtBERT due to its superior performance in catalytic function prediction tasks. The AIDO.Protein-16B model utilizes a transformer encoder architecture enhanced with sparse Mixture-of-Experts (MoE) layers, implementing a top-2 routing mechanism for expert activation. This MoE architecture is particularly advantageous for accurately analyzing protein catalytic capabilities as it allows specialized expert networks to focus on different functional domains and catalytic motifs within protein sequences. Each expert can capture distinct sequence patterns associated with specific catalytic mechanisms, enabling more precise functional annotation. For a given

protein sequence $S = [s_1, s_2, \ldots, s_l]$, we obtain the sequence representation through average pooling: $\mathbf{h}_{\text{protein}} = \frac{1}{l} \sum_{i=1}^{l} s_i$ where $s_i$ represents the contextualized embedding of the $i$-th amino acid residue. The average pooling operation is essential for creating a fixed-dimensional representation that captures the overall sequence characteristics while maintaining computational efficiency, as it aggregates information from all residues equally and provides a robust global sequence representation that is less sensitive to sequence length variations.

**Molecular structure representation.** To better compare the performance of traditional models and large models as embeddings within the OT framework, we evaluate the state-of-the-art vision foundation model DINOv2(Oquab et al., 2024) alongside SMILES-based models such as Mol-BERT(Fabian et al., 2020) and ChemBERTa(Chithrananda et al., 2020). The molecular embeddings are extracted as: $\mathbf{e} = \text{VM}_\theta(\mathbf{I})$ where $\mathbf{I}$ represents the images or SMILES codes. This custom architecture is designed to capture both topological structural information (bond patterns, functional groups) and chemical properties (aromaticity, charge distribution) that are crucial for understanding catalytic mechanisms. For proteins associated with multiple molecular substrates, we compute the aggregate molecular representation through average pooling: $\mathbf{h}_{\text{molecule}} = \frac{1}{n} \sum_{j=1}^{n} \mathbf{e}_j$.

These three modalities capture complementary aspects of protein functionality—relational context from knowledge graphs, sequential patterns from amino acid sequences, and structural cues from molecular substrates. To leverage their synergy for catalytic prediction, we employ optimal transport to joint KG, sequence, and molecular evidence into three transport plans over protein–substrate candidates, yielding OT-based relevance scores for injecting structured catalytic evidence into the LLM.

## 3.2 Neural Optimal Transport

For each modality, we use neural optimal transport networksKorotin et al. (2023) to characterize the relationship between proteins and enzyme substrates. Let $P$ and $Q$ denote the distributions of protein embeddings and substrate embeddings in their respective embedding spaces, i.e., on $X \subset \mathbb{R}^{d_p}$ and $Y \subset \mathbb{R}^{d_s}$. Correspondingly, samples $x \sim P$ and $y \sim Q$ represent vectorized protein and substrate representations, respectively. The standard formulation of the optimal transport cost is: $\text{Cost}(P, Q) \stackrel{\text{def}}{=} \inf_{T_\# P = Q} \int_X c(x, T(x)) \, dP(x)$, where the mass is indivisible. When the equation achieves its minimum value, we obtain the optimal transport map $T^*$. However, the indivisibility constraint prevents it from satisfying the one-to-many relationship required for protein-to-substrate mapping. Therefore, we adopt the relaxed formulation proposed by Kantorovitch (1958):

$$\text{Cost}(P, Q) \stackrel{\text{def}}{=} \inf_{\pi \in \Pi(P,Q)} \int_{X \times Y} c(x, y) d\pi(x, y), \tag{2}$$

where $\pi$ represents the transport plan. The optimal transport plan, denoted as $\pi^*$. Here we introduce potential functions and constraints to obtain the dual formulation of the optimal transport problem. By incorporating the potential functions $\phi : X \to \mathbb{R}^z$ and $\psi : Y \to \mathbb{R}^z$, the Kantorovich dual problem can be expressed as:

$$\sup_{\phi, \psi} \left\{ \int_X \phi(x) dP(x) + \int_Y \psi(y) dQ(y) : \phi(x) + \psi(y) \leq c(x, y), \forall x \in X, y \in Y \right\}.$$

This constraint ensures that the sum of potential functions at any pair of points does not exceed the transportation cost between them. Subsequently, we utilize the C-transform to reformulate the dual problem. The C-transform of function $\phi$ is defined as:

$$\phi^c(y) = \inf_{x \in X} \{ c(x, y) - \phi(x) \}.$$

By applying this transformation, we can eliminate the variable $\psi$ and rewrite the dual problem in its simplified form:

$$\sup_\phi \left\{ \int_X \phi(x) dP(x) + \int_Y \phi^c(y) dQ(y) \right\}.$$

This reformulation reduces the optimization from a two-function problem $(\phi, \psi)$ to a single-function problem over $\phi$, while the C-transform $\phi^c$ automatically ensures satisfaction of the dual constraints.

In our problem, we use MLPs to model the potential functions $\phi$. This transformation is fundamental in optimal transport theory as it establishes the connection between the primal and dual problems through the geometric structure of the cost function.

### 3.3 RETRIEVAL-GENERATION PROCESS

Given a query $Q$ asking about the catalytic relationship between a certain protein, we extract the protein entity from this query as $x$. In our retrieval and filtering process, we simultaneously considered amino acid sequence information of proteins, catalytic molecular formula information, and graph structure information from the biological domain knowledge graph.

After training the neural OT model, we obtain the optimal transport map $T_\theta^*$ and potential function $f_\omega^*$ that capture the underlying catalytic relationships between proteins and substrates. If protein $x$ can catalyze substrate $y$, then the learned transport map should be able to "transport" the protein embedding $x$ to a location in embedding space that is close to the substrate embedding $y$. It can be formulated as

$$P(x \text{ catalyzes } y) = \sigma\left(-\text{dist}(y, T_\theta^*(x, y))\right),$$

where $\sigma(\cdot)$ is the sigmoid function that maps the negative distance to a probability in $[0, 1]$, and $\text{dist}(\cdot, \cdot)$ is a distance metric in the embedding space. We then calculate the cosine similarity between proteins and select the five most similar ones:

$$\text{ProtRenList}(p) = \text{Top}_5(-\text{dist}(y, T_\theta^*(x, y))).$$

In this way, we obtain catalytic molecules of protein entities similar to the target protein entity, retrieving a list of candidate molecular formulas through the distance. Then we crawl the neighbor information of these entities through knowledge graphs to construct context prompts, which are then fed to large language models to generate explanatory information for catalytic predictions. In summary, we extract the rich semantic information in the knowledge graph to extract multidimensional context information for each candidate molecule: a) molecular structural characteristics and chemical properties; b) related biochemical reaction pathways and mechanisms; c) interaction network relationships with other proteins. Our proposed method not only improves prediction accuracy but also provides detailed biological explanations for the results, helping researchers understand the underlying mechanisms and offering valuable guidance for subsequent experimental design.

## 4 EXPERIMENTS

We conduct comprehensive experiments to compare the performance of our method with existing approaches. In this chapter, we will provide detailed descriptions of the datasets we used, the baseline methods for comparison, and the ablation experiments.

**Biological knowledge graph.** We utilize a large-scale biomedical knowledge graph that contains 182,296 triples with 30,294 unique protein entities (subjects) and 18,581 molecular entities (objects) connected through 21 distinct relation types. This knowledge graph encodes rich biochemical relationships, including protein–gene associations, family hierarchies, functional categories, catalytic molecular identifiers, and molecular formulae. All entities are represented using standardized identifiers to ensure consistency across data sources. To complement the graph information, we integrate structural representations of catalytic molecules from the ChEBI database[2] and protein sequence information from UniProt[3]. These additional resources provide molecular structure images and amino acid sequences corresponding to each protein identifier, which are incorporated into our multi-modal representation learning framework.

**Dataset splits.** We partition the catalytic triples extracted from the knowledge graph into training and testing sets using a stratified 90:10 split, ensuring balanced coverage of catalytic relationship types across subsets. This strategy preserves the distributional characteristics of the full dataset, while providing sufficient samples for model optimization and reliable evaluation of generalization performance.

---

[2] https://www.ebi.ac.uk/chebi/
[3] https://www.uniprot.org/

Given the class imbalance commonly observed in biological interaction prediction tasks, we apply a controlled negative sampling procedure. Negative instances are generated at a positive-to-negative ratio of 1:1.5, a setting determined empirically through preliminary experiments to provide robust supervision without overwhelming the positive signal. To prevent data leakage, protein–substrate pairs appearing in the training set are excluded from the test set, regardless of relationship labels. Furthermore, all negative samples are cross-checked against external databases to reduce the risk of inadvertently including unannotated true interactions. This rigorous partitioning and sampling protocol establishes a reliable foundation for training and evaluation of our optimal transport–enhanced catalytic prediction framework.

**Evaluation Metrics.** Following standard practice in knowledge graph embedding (KGE) research, we adopt rank-based metrics including Hits@$k$ (H@$k$, with $k \in \{1, 3, 5\}$) and Mean Reciprocal Rank (MRR) to evaluate model performance. For a given query $(h, r, ?)$ or $(?, r, t)$, the model generates a ranked list of candidate entities, and the rank of the true entity is recorded. H@1 measures the proportion of queries where the correct answer is ranked first, while H@3 and H@5 extend this evaluation to the top three and top five predictions, respectively.

Furthermore, to assess interpretability, we evaluate 50 sampled prediction explanations through expert scoring. Detailed case studies and evaluation results are provided in the Appendix.

## 4.1 BASELINES

In our study, we conducted a comprehensive evaluation by comparing our novel approach with traditional graph neural networks and state-of-the-art RAG methods. This multifaceted comparison allowed us to assess the effectiveness of our model in different paradigms in the computational biology domain.

For graph-based baselines, we selected TransE (Bordes et al., 2013), GNN (Wang et al., 2022), GAT (Elnaggar et al., 2021) and RGCN (Schlichtkrull et al., 2017) as our baselines. These approaches have demonstrated considerable success in capturing complex relationships within structured data. TransE is one of the most influential knowledge graph embedding methods, which models relationships as translation operations in entity embedding space, making the head entity plus the relationship vector approximate the tail entity, thereby effectively representing entities and relationships in a low-dimensional continuous space. We pretrained both the GNN and GAT models using self-supervised learning techniques to effectively capture the inherent patterns and relationships within the biological knowledge graph. This methodology allowed graph-based approaches to learn rich representations of the underlying biological network structure while focusing specifically on catalytic interactions of interest. As an extension of traditional GCN, RGCN introduces relation-specific transformations, enabling it to simultaneously capture node features and different types of relationship information, making it particularly suitable for modeling diverse catalytic relationships and interaction mechanisms between proteins and molecules.

To represent the cutting edge of retrieval-augmented methods, we selected BioRAG (Wang et al., 2024), KG-RAG (Sanmartin, 2024) and BioMedRAG (Li et al., 2024) as our RAG baselines. BioRAG is specifically designed for the biomedical domain, with carefully optimized retrieval mechanisms for scientific literature, genomic databases, and protein information repositories. This system employs deep learning-driven retrievers capable of understanding complex biomedical query intentions and extracting relevant information from large-scale biomedical corpora. KG-RAG is an innovative knowledge graph-enhanced retrieval generation system that combines traditional text retrieval with structured knowledge graphs. This method utilizes the rich semantic relationships in biomedical knowledge graphs to guide the retrieval process. BioMedRAG focuses on the intersection of medicine and biology, providing enhanced support for clinical and biological research. This system integrates medical literature, clinical guidelines, drug databases, and genomic information to build a multi-source retrieval framework.

However, despite their respective advantages, these systems primarily rely on textual information and fail to fully utilize the rich multimodal data in modern biological research. Our CataRAG method aims to address this limitation by integrating protein sequences, molecular structure images, and knowledge graph information to provide comprehensive catalytic relationship prediction.

## 4.2 MAIN RESULTS

The experimental results presented in Table 1 demonstrate the performance comparison of different retrieval models in multiple evaluation metrics. Our proposed CataRAG Given that the molecular modeling field features two highly effective approaches—one based on SMILES encoding and another based on vision large models—we conducted experiments using three different encoder architectures: MolBERT, ChemBERTa, and DINOv2. The results demonstrate that the choice of molecular encoder significantly affects model performance across different evaluation metrics. CataRAG with DINOv2 achieves the highest performance at H@1 (0.807±0.014), while CataRAG with ChemBERTa shows superior results at H@5 (0.913±0.015). Notably, MolBERT-based CataRAG exhibits the most consistent performance with relatively balanced scores across all metrics, whereas DINOv2 shows higher variance in MRR (0.438±0.083), suggesting potential instability in certain prediction scenarios. Furthermore, we compared our method against Neural OT, which employs optimal transport-based ranking algorithms as a RAG module to enhance LLMs performance. Neural OT demonstrated strong competitive performance, achieving H@1, H@3, and H@5 scores of 0.761±0.023, 0.794±0.015, and 0.801±0.023 respectively, with an MRR of 0.375±0.064. Despite the competitive performance of Neural OT, our CataRAG method still significantly outperformed it across all evaluation metrics, particularly showing notable advantages in H@1 (0.807 vs 0.761) and MRR (0.445 vs 0.375).

In comparison, traditional graph-based methods like GNN and GAT showed moderate performance, with GAT (0.571, 0.648, 0.653 for H@1, H@3, H@5) performing better than GNN (0.431, 0.533, 0.534). The poor performance of GNN can be attributed to its failure to incorporate protein amino acid sequence information, relying solely on semantic information from knowledge graphs, which proves insufficient for accurate prediction. Furthermore, GNN cannot make predictions for enzyme-substrate pairs that are absent from the knowledge graph, significantly limiting its applicability. The TransE baseline demonstrated the weakest performance among all tested models, with consistently low scores across all metrics (0.231, 0.472, 0.524) and the lowest MRR (0.221).

Among retrieval-augmented methods, BiomedRAG performed relatively well with an MRR of 0.357±0.059, followed by KG-RAG (MRR of 0.299±0.023), while BioRAG exhibited the weakest performance among all tested models, with H@1, H@3, and H@5 of 0.354±0.132, 0.362±0.023, and 0.356±0.043 respectively, and an MRR of just 0.180±0.054. The suboptimal performance of BioRAG stems from its reliance on text-based retrieval materials for large language model predictions, which also neglects crucial protein amino acid sequence structural information. Additionally, the lengthy contextual information fails to provide precise guidance for the large language model's predictions, contributing to its inferior performance. These results convincingly demonstrate the superiority of our proposed CataRAG method in biomedical catalytic relationship prediction tasks.

Table 1: Performance comparison of different baseline models. Baseline methods can be divided into two categories: traditional graph methods and LLMs-based RAG methods.

| Model | H@1 | H@3 | H@5 | MRR |
|---|---|---|---|---|
| TransE | 0.231±0.021 | 0.472±0.034 | 0.524±0.012 | 0.221±0.038 |
| GNN | 0.431±0.013 | 0.533±0.011 | 0.534±0.010 | 0.238±0.014 |
| GAT | 0.571±0.032 | 0.648±0.023 | 0.653±0.022 | 0.304±0.012 |
| RGCN | 0.601±0.045 | 0.760±0.066 | 0.842±0.042 | 0.262±0.053 |
| KG-RAG | 0.528±0.067 | 0.545±0.087 | 0.569±0.076 | 0.299±0.023 |
| BioRAG | 0.354±0.132 | 0.362±0.023 | 0.356±0.043 | 0.180±0.054 |
| BiomedRAG | 0.580±0.087 | 0.598±0.019 | 0.622±0.012 | 0.357±0.059 |
| Neural OT | 0.761±0.023 | 0.794±0.015 | 0.801±0.023 | 0.375±0.064 |
| CataRAG(Molbert) | 0.789±0.049 | 0.852±0.012 | 0.901±0.026 | 0.451±0.032 |
| CataRAG(chemBERTa) | 0.785±0.021 | 0.849±0.024 | 0.913±0.015 | 0.448±0.015 |
| CataRAG(Dinov2) | **0.807±0.014** | **0.821±0.034** | **0.875±0.025** | **0.438±0.083** |

## 4.3 ABLATION STUDY

To better understand the contribution of different components in our model architecture, we conducted an ablation study that compared the three encoder modules of our approach. Table 2 shows the performance of different modules in multiple evaluation metrics.

We attempted to remove the retrieval augmented module, using only ChatGPT-4o to answer our constructed catalysis questions and calculate metrics. Without accurate biological domain information guidance, large models showed poor accuracy for open-ended questions (H@1 0.325, H@3 0.341, H@5 0.045). The GNN module achieved performance with H@1 of 0.532, H@3 of 0.563, and H@5 of 0.581, resulting in an MRR of 0.287. This indicates that GNN provides a reasonable foundation for capturing structural relationships in the data. The VLM module has a consistent performance improvement across all metrics. The H@1 increased to 0.612, while H@3 and H@5 improved to 0.653 and 0.681 respectively. The MRR also showed a notable increase to 0.324, representing a 0.129 relative improvement over the GNN module. These results demonstrate the value of molecular structural information in enhancing catalytic relationship prediction. MolBERT and ChemBERTa achieved performance comparable to our VLM method, while performing slightly lower than VLM across all three metrics.

The sequence module achieved the best performance across all metrics, with substantial gains over both the GNN module and the image-only module. Specifically, the sequence module reached a H@1 of 0.673, H@3 of 0.697, and H@5 of 0.709, with an MRR of 0.409. This represents a 0.265 relative improvement in H@1 and a 0.425 increase in MRR compared to the GNN module. This indicates that protein amino acid sequence modeling plays a crucial role in the final results.

Combining the experimental results of CataRAG, we can see that when we integrate the three modules together, the prediction H@1 improves from the best baseline of 0.673 to 0.807, indicating that our selected three modules possess certain complementarity at the embedding level and can achieve mutually enhancing effects.

Table 2: Ablation study results demonstrating the contribution of different components.

| Model | H@1 | H@3 | H@5 | MRR |
|---|---|---|---|---|
| ChatGPT-4o | 0.325±0.023 | 0.341±0.034 | 0.361±0.045 | 0.176±0.023 |
| GNN module | 0.532±0.029 | 0.563±0.074 | 0.581±0.047 | 0.287±0.095 |
| VLM module | 0.612±0.031 | 0.653±0.054 | 0.681±0.021 | 0.324±0.015 |
| Molbert | 0.608±0.013 | 0.642±0.032 | 0.686±0.019 | 0.322±0.010 |
| ChemBERTa | 0.609±0.014 | 0.643±0.051 | 0.673±0.023 | 0.323±0.012 |
| Sequence module | 0.673±0.032 | 0.697±0.023 | 0.709±0.012 | 0.409±0.048 |

## 5 CONCLUSION

We introduced **CataRAG**, a retrieval-augmented framework that enables large language models to incorporate structured biochemical knowledge from protein sequences, molecular structures, and biological knowledge graphs. By coupling multimodal retrieval with an optimal transport–based fusion mechanism, the framework reduces the reliance on expensive pretraining of domain-specific language models and improves the accuracy and interpretability of catalytic interaction prediction. Our experiments show that CataRAG consistently outperforms both general-purpose LLMs and existing graph-based baselines, highlighting the value of combining symbolic knowledge resources with modern language models in biochemical reasoning.

Looking forward, an important next step is to develop a self-improving loop in which newly predicted interactions are fed back into the knowledge graph, allowing the system to refine its retrieval space as more biochemical evidence accumulates. Such an adaptive mechanism would be especially valuable in biomedicine, where new protein functions and molecular interactions are discovered at a rapid pace.

**Limitations** CataRAG depends on curated corpora such as UniProt and biological knowledge graphs, which may not always be available or complete for every domain. Large-scale retrieval from these resources can also raise efficiency concerns for real-time applications. Finally, the overall performance of the pipeline remains constrained by the capacity of pretrained protein language models and molecular encoders, which suggests that future advances in foundation models for biology could directly benefit our framework.

ETHICS STATEMENT

This work focuses on computational prediction of protein-enzyme substrate catalysis and does not involve human subjects or raise ethical concerns. All protein and enzyme data used in this study are obtained from publicly available databases and do not contain sensitive personal information.

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

## A  IMPLEMENTATION DETIALS

We implemented our experiments using PyTorch and conducted them on an NVIDIA GPU.

When training our self-supervised GNN, we implemented a comprehensive configuration to ensure optimal performance. For optimization, we selected the Adam optimizer due to its adaptive learning rate capabilities and efficient handling of sparse gradients. We configured the optimizer with an initial learning rate of 0.001, which provided a good balance between convergence speed and stability during the training process. To mitigate overfitting and improve generalization, we applied a weight decay regularization of 5e-4.The training regimen consisted of 300 epochs, which provided sufficient iterations for the model to learn meaningful representations from the graph data. We carefully monitored the validation metrics throughout this extended training period to ensure the model was learning effectively without overfitting. Additionally, we set the batch size to 8, which was determined to be optimal for our hardware configuration and dataset characteristics, balancing computational efficiency with statistical stability in gradient updates.This configuration resulted in a robust GNN model that effectively learned self-supervised representations from the graph data.

For the molecular formula structures, We obtained protein catalytic molecular formulas through a web crawler program from UniProt. We utilize a VLM - specifically - Dinov2 - which enables us to fully capture and represent the structural characteristics of the molecular formulas. For protein representation, we access protein amino acid sequences through the UniProt database and model these sequences using the AIDO.Protein-16B protein language model, which effectively captures the sequential patterns and functional properties of proteins. This dual representation approach allows us to effectively model both the molecular structures and protein sequences, providing a robust foundation for our subsequent analyses and predictions.

## B  DATA STATISTICS

This study conducts a comprehensive analysis of a large biomedical knowledge graph dataset, containing 182,296 triples that involve 30,294 unique source entities (subject) and 18,581 target entities (object), interconnected through 21 different types of relationships.

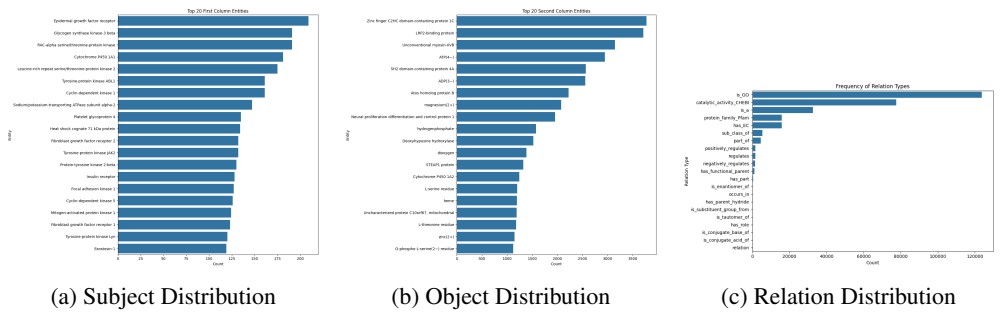

(a) Subject Distribution  (b) Object Distribution  (c) Relation Distribution

Figure 2: Data statistics visualization

## B.1 ENTITY ANALYSIS

The analysis shows that subject entities in the dataset primarily consist of various proteins and enzymes, with Epidermal Growth Factor Receptor (EGFR) appearing most frequently at 209 records. Other high-frequency source entities include Glycogen Synthase Kinase-$3\beta$ (GSK-$3\beta$) and RAC-$\alpha$ Serine/Threonine Protein Kinase, both with 191 records. These high-frequency entities are predominantly molecules that play key roles in cell signaling and metabolic regulation.

Regarding object entities, Zinc Finger C2HC Domain-containing Protein 1C and LRP2-binding Protein are the two most frequent proteins, appearing 3,775 and 3,709 times respectively. Energy-related molecules such as ATP(4-) and ADP(3-) also rank in the top ten, with 2,949 and 2,557 records respectively, highlighting the central role of energy metabolism in biological systems.

## B.2 RELATION ANALYSIS

The distribution of relationship types reveals that "is_GO" is the most predominant relationship type, accounting for 43.8% triples of the total. This indicates that the knowledge graph is tightly integrated with Gene Ontology (GO) terms, providing a rich context for protein function annotation. The second most common relationship is "catalytic_activity_CHEBI", covering 27.5% triples, which primarily describes enzymatic reactions and their substrates. Hierarchical relationships such as "is_a" (11.6%) and "sub_class_of" (1.9%) together form the taxonomic backbone of the knowledge graph.

Overall, these statistical findings reveal the rich content and complex structure of the biomedical knowledge graph, highlighting the central role of key proteins, enzymes, and molecules in biological systems, providing a valuable foundation for in-depth research on functional relationships and interaction networks between biomolecules.

## C  CASE STUDY

We have defined the format and prompt for the model's final output as followed. The prompt clearly defines the task objective: to analyze an input protein and a list of candidate molecular formulas, then identify which molecules are most likely to be catalyzed. The instruction section directs the model to select the most probable 1, 3, or 5 molecular formulas (adjustable based on specific needs), and to provide a detailed explanation for each selected formula, including key interactions with the protein's active site, relevant chemical mechanisms, supporting evidence from similar proteins or reactions, and potential challenges or limitations.

The output format is standardized as a structured JSON, requiring the model to provide protein analysis, predictions for each molecular formula (including ranking, formula name, and detailed explanations), and an overall analysis summary. This structured output not only facilitates subsequent automated processing and analysis but also makes the results easy to understand and compare.

702
703

---

**Protein Catalytic Molecule Prediction Prompt**

704
705
706
707

Task: Analyze a protein and identify the most likely catalytic molecular formulas from a provided list.
Input: - Protein query: Protein name - List of candidate molecular formulas: [List of molecular formulas name]
Instructions:
1. Select the top [1/3/5] most likely catalytic molecular formulas.
5. For each selected formula, provide: Detailed explanation of why this molecule is likely to be catalyzed, including: * Key interactions with the protein's active site * Relevant chemical mechanisms * Supporting evidence from similar proteins or reactions * Potential challenges or limitations
Output Format:
Return your analysis as a structured JSON with the following format:
{
"protein_analysis": {
"protein_name": "string",
},
"predictions": [ {
"rank": 1,
"molecular_formula": "string",
"explanation": "string"
},
// Additional predictions as needed
],
"summary": "string" // Overall analysis summary
}

---

Regarding the accuracy of catalytic molecule predictions, a detailed analysis and explanation has been provided in Section 4. In this section, we analyze the explanatory information generated by our model. Our analysis focuses exclusively on the explanations provided for correctly predicted protein entities.

To evaluate the correctness of these explanations, we recruited three PhD students from the biological sciences field. They assessed the accuracy of the protein reaction explanations using a binary scoring system: 1 for correct explanations and 0 for incorrect explanations. Our results show that the explanations for correctly predicted results achieved an average accuracy rate of 74%. Below is a case example:

---

**Explanation case**

Probable mannose-6-phosphate isomerase GmuF is an enzyme that catalyzes the isomerization of mannose-6-phosphate to fructose-6-phosphate, a crucial step in carbohydrate metabolism. The enzyme likely employs a catalytic mechanism that involves the binding of zinc(2+) ions. Zinc(2+) acts as a cofactor, stabilizing the negative charges that develop during the isomerization process. It facilitates the opening of the sugar ring and the subsequent conversion of the aldose form (mannose-6-phosphate) to the ketose form (fructose-6-phosphate). The presence of zinc(2+) enhances the enzyme's ability to stabilize the transition state, thereby increasing the reaction rate and ensuring efficient catalysis. FAD-dependent monooxygenase janM is an enzyme that utilizes flavin adenine dinucleotide (FAD) as a cofactor to catalyze oxidative reactions. The enzyme operates by binding FAD in its oxidized form (FAD) and subsequently reducing it to FADH2 through the acceptance of electrons. In the presence of molecular oxygen, FADH2 is reoxidized to FAD, generating a reactive oxygen species, typically a peroxide or superoxide. This reactive oxygen species can then facilitate the hydroxylation or epoxidation of substrates, incorporating an oxygen atom into the substrate molecule. The regeneration of FAD allows the enzyme to continuously catalyze successive reactions, making it efficient in oxidative biotransformations.

---

# D   ABLATION STUDY

We also compared the impact of different Protein language models and VLMs on the results. In our pipeline, we selected the best-performing models from these two sections.

## D.1   PROTEIN LANGUAGE MODELS

We compared the following three models: 1. Protein-16B is a specialized protein language model with 16 billion parameters designed to effectively capture protein sequence patterns and structural information for various biological applications. 2. ESM2-15B is a transformer-based evolutionary scale modeling approach developed by Meta AI with 15 billion parameters that learns protein representations from millions of diverse sequences to predict structure and function. 3. xTrimoPGLM-100B is a massive 100-billion-parameter protein language model that integrates multidimensional biological information to achieve comprehensive understanding of protein sequences, structures, and functions across diverse tasks.

Table 3: Ablation study results demonstrating the contribution of different PLMs.

| Model | H@1 | H@3 | H@5 | MRR |
|---|---|---|---|---|
| Protein-16B | **0.807±0.014** | **0.821±0.034** | **0.875±0.025** | **0.438±0.083** |
| ESM2-15B | 0.746±0.023 | 0.753±0.042 | 0.793±0.012 | 0.432±0.015 |
| xTrimoPGLM-100B | 0.748 | - | - | - |

The table 4 presents a comparative analysis of three protein language models: Protein-16B, ESM2-15B, and xTrimoPGLM-100B. Across all evaluation metrics, Protein-16B demonstrates superior performance, with the highest scores highlighted in bold. Specifically, Protein-16B achieves scores of 0.807±0.014, 0.821±0.034, 0.875±0.025, and 0.438±0.083 across the four evaluation criteria.

ESM2-15B shows the lowest performance among the three models, with scores of 0.746±0.023, 0.753±0.042, 0.793±0.012, and 0.432±0.015. Due to the substantial size of xTrimoPGLM-100B (100 billion parameters), our current computational resources are insufficient to utilize this model for generating embeddings, so we directly incorporate the results reported in the paper(Sun et al., 2024).

## D.2   VLMs

We compared the following two models: 1. Dinov2 is a vision-language model developed by OpenAI that divides images into 32×32 pixel patches and uses a 12-layer transformer architecture to create aligned visual and textual representations for diverse cross-modal tasks. 2. ViT-B/32 is a self-supervised vision model developed by Meta AI that learns powerful visual representations without relying on labeled data, using a teacher-student distillation approach and a Vision Transformer architecture to achieve state-of-the-art performance on a wide range of computer vision tasks.

Table 4: Ablation study results demonstrating the contribution of different VLMs.

| Model | H@1 | H@3 | H@5 | MRR |
|---|---|---|---|---|
| Dinov2 | **0.807±0.014** | **0.821±0.034** | **0.875±0.025** | **0.438±0.083** |
| ViT-B/32 | 0.776±0.032 | 0.821±0.031 | 0.871±0.025 | 0.440±0.015 |

The table presents a comparative analysis of two vision models: ViT-B/32 and Dinov2(Oquab et al., 2024). Across all four evaluation metrics, Dinov2 demonstrates superior performance, with its results highlighted in bold. Specifically, Dinov2 achieves scores of 0.807±0.014, 0.821±0.034, 0.875±0.025, and 0.438±0.083 across the four performance criteria.

While ViT-B/32 shows competitive performance with scores of 0.776±0.032, 0.821±0.031, 0.871±0.025, and 0.440±0.015, it falls slightly short of Dinov2 in all metrics. The difference between the two models is relatively small, suggesting that both are effective, though Dinov2 holds a consistent edge across all evaluation measures.

