# OpenReview forum: "CataRAG: LLM-based Enzyme-Substrate Prediction with Multi-modal Optimal Transport Maps"
_ICLR.cc/2026/Conference — Submitted to ICLR 2026_

### Official Review · Reviewer_eQX2 · 2025-10-26

**Soundness:** 3
**Presentation:** 3
**Contribution:** 2
**Rating:** 2
**Confidence:** 4

**Summary:**

This paper proposes a retrieval-augmented biological reasoning framework, which uses the multi-modal optimal transport theory to integrate information from various modalities, and uses the final multi-modal retrieval results to augment the complex biological reasoning of LLMs. Experimental results on the knowledge graph link prediction showcase the superiority of the proposed method.

**Strengths:**

S1. Integrating the multi-modality information within the biological reasoning is a valuable and interesting research problem.

S2. The paper is well-written and easy to follow.

**Weaknesses:**

W1. This paper appears to lack technical contributions. The encoders for each modality rely on existing models, and the multimodal alignment method follows prior work on neural optimal transport networks—specifically, the core optimal transport component itself is not novel. Furthermore, the retrieval-augmented generation framework employed, where retrieved context is fed into a large language model for reasoning, represents a standard paradigm without significant innovation.

W2. The authors assert that the retrieved context is employed to enhance complex biological reasoning. However, in their experiments, model performance is evaluated only on a simple knowledge graph link prediction task, with no demonstration of its ability to handle complex reasoning.

W3. Some important experimental results are not provided, for example, the efficiency analysis of the proposed framework, the detailed analysis of the retrieval performance, the hyper-parameter analysis, the effect of using different knowledge graphs, and the failure case analysis.

**Questions:**

Q1. KGs are known to be incomplete. What will happen if the knowledge graph contains no useful information for a query triple?

---

### Official Review · Reviewer_ZpKS · 2025-10-26

**Soundness:** 2
**Presentation:** 2
**Contribution:** 2
**Rating:** 4
**Confidence:** 3

**Summary:**

This paper introduces CataRAG, a retrieval-augmented generation (RAG) framework that integrates multimodal information, including protein amino acid sequences, molecular substrate structures, and biological knowledge graphs, using multimodal optimal transport to improve enzyme-substrate catalytic interaction prediction. The framework matches protein and substrate representations via neural optimal transport across three distinct embedding modalities and aggregates the retrieval results, supplying relevant structured knowledge to large language models for catalytic reasoning and explanation. Comprehensive experiments benchmark CataRAG against state-of-the-art LLMs, RAG baselines, and graph neural networks, and ablation studies are conducted to examine the contributions of each modality.

**Strengths:**

1. Multi-Modal integration: The work systematically combines protein sequences, molecular graphs (via visual or SMILES-based encoders), and knowledge graph embeddings for enzyme-substrate prediction, addressing limitations of unimodal approaches.
2. Clear problem motivation: The paper clearly motivates the need for richer, evidence-grounded catalytic reasoning in biological link prediction, where classical LLMs and unimodal models fall short, particularly due to hallucination and lack of structured biological grounding.
3. Comprehensive experiments: The experimental evaluation benchmarks CataRAG against strong and diverse baselines—including multiple knowledge-graph methods and RAG-based systems. Ablation studies dissect the impact of each modality and backbone.
4. Quantitative advancement: The main results table shows state-of-the-art performance in MRR and Hits@k metrics on the catalytic interaction prediction task, with DINOv2+MolBERT-based variants of CataRAG outperforming all baselines. Figure 1 provides a helpful overview of the system pipeline and justifies design choices for multi-modal fusion and retrieval.

**Weaknesses:**

1. Unclear OT fusion mechanism: One of the core contributions is the use of multi-modal OT. However, its specific implementation is poorly described. Figure 1 and its caption  imply that a separate OT network is designed for each modality to map to the enzyme substrate embedding space. Yet, Methods section 3.2  describes a general neural OT network without specifying how these three independent information streams are processed. How the "transport plans" or distances from these three OT paths are finally combined via "weighted aggregation"  is also not detailed with a formula or clear description.
2. The paper does not clearly state the role of the LLM in the final prediction (i.e., the metrics in Table 1). Is the LLM merely used to generate explanations, or does it re-rank the top-k results returned by the retriever? If it's the former, then the results in Table 1 do not fully represent the predictive performance of the entire RAG framework, but only that of its retrieval component.
3. Insufficient ablation study. The ablation study lacks pairwise combinations (e.g., GNN + Sequence, GNN + VLM, Sequence + VLM). Without these intermediate steps, it is difficult to judge which pair of modules contributes most to the significant performance jump (from 0.673 to 0.807) and weakens the conclusion about the complementarity of the three modalities.
4. Fairness of baseline comparisons. The authors claim CataRAG's main advantage is fusing multi-modal information. However, the chosen baselines are either graph-based methods or RAG methods that the authors state rely primarily on text. A simple multi-modal baseline is missing. For example, a baseline that simply concatenates or averages the embeddings from the three modalities and then uses a standard similarity metric (like cosine similarity) for retrieval. Without this comparison, it is difficult to convince a reviewer that CataRAG's performance gain is due to the complex OT fusion mechanism rather than simply introducing multi-modal information.

**Questions:**

1. Regarding the OT fusion mechanism: Did you train three independent OT maps (one for each of the GNN, VLM, and Sequence modalities), or did you fuse the three representations first and then train a single, unified OT map? Can you provide a more explicit mathematical formula or algorithmic description of how you combine the information from the three paths via "weighted aggregation"  to produce the final similarity score and ranking?
2. Regarding the RAG evaluation: Are the H@k and MRR metrics reported in Table 1  the (A) direct ranking output from the OT retriever, or (B) the final prediction results after the LLM (like ChatGPT) analyzed the retrieved context and re-ranked or selected the candidates?
3. Regarding the necessity of OT: Compared to your complex OT fusion method, how does a simpler multi-modal fusion baseline perform (e.g., concatenating the three embeddings and training an MLP for similarity scoring)? This is crucial for demonstrating the true contribution of the OT mechanism.
4. Regarding the ablation study: Can you provide the ablation results for pairwise combinations, such as GNN+Sequence and Sequence+VLM? This would provide stronger evidence for the necessity of fusing all three modalities.
5. Regarding the "Neural OT" baseline: Your "Neural OT" baseline  also appears to use OT for ranking. What is the key innovation of your OT application in CataRAG compared to this baseline? Is it simply that CataRAG handles multi-modal inputs?

---

### Official Review · Reviewer_xZtv · 2025-11-01

**Soundness:** 2
**Presentation:** 3
**Contribution:** 1
**Rating:** 2
**Confidence:** 4

**Summary:**

This paper notices that existing works on protein language models are unimodal and lack complex reasoning ability. To mitigate these problems, this paper aims to integrate three modalities for a comprehensive reasoning, i.e., amino acid sequences, molecular structures of enzymatic substrates, and biological knowledge graphs. Specifically, this paper designs a retrieval-augmented method and proposes a multi-modal optimal transport. Experiments verify the effectiveness of the proposed method.

**Strengths:**

1. Overall, the paper is well written, with figures as visual illustrations. The Introduction section clearly explains the motivation behind the method. It also makes a comparison to existing methods and identifies their drawbacks.

2. Using the recent retrieval-augmented method to enhance the reasoning ability of biological language models is interesting. The proposed multi-modal optimal transport is also novel to me.

3. Experiments contain both hit rate and MRR as evaluation metrics, which comprehensively verify the effectiveness of the proposed method.

**Weaknesses:**

1. Though this paper proposes an interesting method, it misses to mention and compare to a highly related existing work [1], which uses retrieval-augmented method to integrate knowledge graph into proteins' amino acid sequences for a multi-modal representation learning.

[1] Zhang, J., Zhang, D. C., Liang, S., Li, Z., Ying, R., & Shao, J. Retrieval-Augmented Language Model for Knowledge-aware Protein Encoding. In Forty-second International Conference on Machine Learning.

2. There is a lack of ablation study to test the effect of different knowledge graphs, since the proposed method heavily relies on the quality of knowledge graph.

3. Authors are suggested to provide a formal algorithm to show the learning process of the model.

**Questions:**

N/A

---

### Meta-Review · Area_Chair_ztc9 · 2026-01-07

**Summary:**

After careful consideration of the paper and the points raised by the reviewers, it is clear that the paper, in its current form, does not meet ICLR standards. While the paper addresses the critical issue of enzyme-substrate interaction prediction using a multi-modal approach, reviewers propose significant and overlapping concerns regarding method clarity, experimental rigor, and the positioning of the paper's contribution.

**Reviewer Concerns:**

- Lack of Clarity in the Core Technical Method (Reviewer ZpKS): The reviewer finds the proposed method confusing.
- Insufficient and Unfair Experimental Comparisons (Reviewers xZtv, ZpKS, eQX2): The experimental setup misses important baseline comparisons to demonstrate the effectiveness of the proposed method under fair conditions.
- The current ablation study is not thorough enough to validate the central claim about the complementarity of the three modalities (Reviewers xZtv, ZpKS).
- The paper lacks significant technical contributions (Reviewer eQX2).

**Reviewer Scores:**

The authors did not participate in the rebuttal discussion. As a result, reviewers are likely to maintain their original scores.

---

### Decision · Program_Chairs · 2026-01-26

Reject